# Effect of Application of Different Activation Media on Fertilization and Embryo Survival of Northern Pike, (*Esox lucius*) under Hatchery Conditions

**DOI:** 10.3390/ani12081022

**Published:** 2022-04-14

**Authors:** Marek J. Łuczyński, Joanna Nowosad, Joanna Łuczyńska, Dariusz Kucharczyk

**Affiliations:** 1Department of Ichthyology, Hydrobiology and Ecology of Waters, The Stanislaw Sakowicz Inland Fisheries Institute, 10-719 Olsztyn, Poland; 2Department of Veterinary Prevention and Feed Hygiene, Warmia and Mazury University, 10-719 Olsztyn, Poland; nowosad.joanna@gmail.com; 3ChemProf, 11-041 Olsztyn, Poland; 4Department of Commodity and Food Analysis, Warmia and Mazury University, 10-726 Olsztyn, Poland; jlucz@uwm.edu.pl; 5Department of Ichthyology and Aquaculture, Warmia and Mazury University, 10-701 Olsztyn, Poland

**Keywords:** insemination, controlled reproduction, gamete management, wild stock, aquaculture

## Abstract

**Simple Summary:**

Northern pike is one of the most important freshwater species that is produced mainly for restocking. It is economically and ecologically valuable and an important species in recreational fisheries as a top predator in fresh waters. Although the artificial reproduction of this species seems easy, it still poses many difficulties. This is why research is constantly being conducted to improve artificial reproduction, the short-term storage of gametes, and fertilization techniques. The aim of this study was to use various activation media to increase fertilization rates in this species. The results obtained indicate that the use of selected activation media in field hatcheries can significantly increase the reproductive efficiency of northern pike.

**Abstract:**

One of the finfish species that European and North American breeders are most interested in is the northern pike, *Esox lucius*. Artificial reproduction and the production of viable larvae has a huge impact on further culture. The quality of stripped gametes is highly variable. Therefore, it is important to use gametes with maximum efficiency, which has a direct impact on the amount of stocking material produced and therefore on the economics of production. The aim of this study was to compare northern pike fertilization efficiency, expressed as the survival rate of embryos until hatching. In the first experiment, the highest percentage of hatched embryos was observed in groups of eggs fertilized in a saline diluent prepared with deionized water (after reverse osmosis: group D), at 61.2% and 56.5% in the NaCl5-D and NaCl6-D groups, respectively. The highest percentage of hatched embryos in the second experiment was observed in the egg groups activated with Woynarovich solution (V) at 75.5% and 74.7% for V-D (D—deionized water) and V-T (T—tap water), respectively. In all cases, preparing the activation medium using T versus D water resulted in lower fertilization percentages and lower percentages of hatched larvae. At the same time, two variants (V and B1—Billard solution) were tested during mass spawning in three hatcheries using hatchery water (tap water). The results showed that repeatability was the highest when using activation medium B1.

## 1. Introduction

The increasing demand for food of aquatic origin [1] drives the constant development of artificial fish reproduction technologies [2,3,4,5,6], the modification of fertilization methods [7,8,9,10], genomic manipulation [11,12,13], and egg incubation [14], which facilitate attainment of high-quality larvae for further rearing stages [15,16,17]. One of the finfish species that breeders in Europe and North America are most interested in is the northern pike, *Esox lucius* [18,19,20,21]. Commercial-sized fish are mainly produced in ponds or in the wild. In both cases, the material for further production is obtained mainly from field hatcheries [20,22]. Artificial northern pike reproduction differs from that of many other freshwater fishes because the biological characteristics of this species require using spawners captured from the wild that are held in lakes or ponds until they reach full sexual maturation before they are transported to hatcheries. The quality of stripped gametes is highly variable (e.g., [18,19,22]). Usually, the males produce relatively small amounts of milt, and at the end of the spawning season in northeastern Poland (which is usually in early May), only very small amounts are observed. Hormonal stimulation of spawning remains ineffective in this species [18,19]. Therefore, it is important to use the gametes that are obtained with maximum efficiency, which has a direct impact on the amount of stocking material produced and therefore on the economic costs of its production [23].

From the point of view of gamete management, water is not the best activation medium [24]. Water causes sperm deformations [25] and induces an exceptionally fast cortical response in oocytes [26], which causes rapid micropyle closure, lowering the percentage of fertilized eggs. Sperm are only briefly motile in water [7,10,27,28] and not for more than 70 s in northern pike [29]. Fish sperm are activated by decreases in osmotic pressure that occur after milt is released into the water and diluted [30]. The duration of sperm motility is also influenced by temperature, ions, and pH [24,31,32,33]. Improving existing methods of fertilization in species such as burbot (*Lota lota*), asp (*Leuciscus aspius*), ide (*Leuciscus idus*), common dace (*Leuciscus leuciscus*), common barbel (*Barbus barbus*), and African catfish (*Clarias gariepinus*) resulted in significant improvement in fertilization and embryo survival, reduced the percentage of deformed embryos, and rendered fertilization less labor-intensive [7,9,10,27,28,34,35,36]. Quick, portable methods for assessing gamete quality are being developed that will help to eliminate low-quality gametes from mass spawning. Combining several methods into a working artificial reproduction protocol that includes gamete quality assessment and fertilization optimization can provide many benefits to field hatcheries [28,37,38,39]. Various activation media were tested in pilot studies conducted separately for each medium. Because extensive comparative studies of many media have not been performed simultaneously on the same batch of northern pike gametes, the aim of the present study was to compare fertilization efficiency in northern pike, expressed as embryo survival rates until hatching. The detailed aim of experiments was to evaluate the effect on fertilization rates of either tap (T) or deionized (D) water used to prepare diluents.

## 2. Materials and Methods

### 2.1. Ethical Content

All procedures on the fish were conducted in accordance with the regulations set forth in the Act on the Protection of Animals Used for Scientific or Educational Purposes (Polish Journal of Laws of 2015, item 266, of 15 January 2015 and European Union 2010/63/UE). The managers of fishing districts obtained the consent of the respective marshal’s offices to catch northern pike spawners during the closed season (1 March–30 April) for use in artificial reproduction to produce stocking material. There were no endangered or protected species involved in the study. The authors of the study (M.J.Ł., J.N., and D.K.) hold a certificate of professional competence for designing experiments and experimental projects.

### 2.2. Experimental Design: General Content

All gametes were collected from wild northern pike spawners held at Pasłęk Fish Hatchery (Olsztyn, northeastern Poland) according to procedures described by Szabo [18], Babiak et al. [40], and Cejko et al. [41]. Wild mature spawners were caught with traps and gill nets and were transported live (in water) to the hatchery. Gendered spawners were separated into tanks of 2–4 m^3^ and kept indoors [42]. The tanks were covered to prevent fish from jumping out. Water temperature and photoperiods were close to natural conditions. Fish density did not exceed 25 kg m^−3^. Dozens of spawners were caught at one time. Spawners were first checked the day after catching and then every 2–3 consecutive days. The weight of the caught spawners ranged from 1 kg to more than 10 kg, but medium-sized fish were selected for the study as gamete donors. Gametes were stripped by gently massaging the spawners’ abdomens. Until the experiment proper, all eggs were kept in small plastic containers at a temperature of 10–12 °C. Milt was kept in plastic syringes on crushed ice at a temperature of 2–4 °C [43]. The storage time of the gametes, counted from the start of stripping to fertilization, did not exceed 3 h.

Gametes for mass fertilization were collected at hatcheries located in northeastern Poland near Olsztyn: Pasłęk (88 km from Olsztyn), Pasym (32 km), and Sterławki (92 km). At all three hatcheries, eggs were stripped from a minimum of 10 females (2–6 kg each), whereas milt containing no impurities was stripped from at least 10 individuals (1–3 kg each).

Two separate experiments were conducted on two consecutive days. Two variants of each diluent were prepared: one with tap (TW) and one with deionized (DW) water (Table 1 and Table 2). The parameters of T water were: conductivity (25 °C), 516 μS; pH = 7.43; total hardness, CaCO_3_ 255 mg L^−1^; Na^+^, 10.8 mg L^−1^; K^+^, 4.9 mg L^−1^; Fe, 12 µg L^−1^; and Mn, 26 µg L^−1^. Conductivity was measured with a Hanna HI 98129 apparatus (Hanna Instruments, Nusfalau, Romania), and pH was determined by an Edge HI2020-02 pH meter (Hanna Instruments, Nusfalau, Romania).

#### 2.2.1. Design of Experiment No. 1

In experiment 1, eggs were collected from seven females, and milt was collected rom from five males. Small batches of eggs were examined microscopically (Olympus SZX16, Olympus, Tokyo, Japan), and eggs from five females were used in the experiment. The eggs were pooled in equal proportions from each female. Milt was pooled, and only batches with sperm motility >80% were used [21,22,35]. The experimental groups were run in triplicate. Each replicate consisted of 3 mL of eggs (approximately 350 eggs), and each batch of eggs was mixed with 30 µL of intact sperm; then, 3 mL of activating media was added (Table 1). The following media were tested: TW (tap water; control group), 0.5% NaCl solution, 0.6% NaCl solution, B1, and V. All test groups were performed on tap (T) and deionized (D) water. Two minutes after gamete activation, each batch of gametes was transferred to separate mini-Weiss incubators (capacity approximately 100 mL). The eggs were incubated until hatching in water at a temperature of 12.0 °C, which is the optimal temperature for northern pike embryonic development [47]. Hatching rate (hatched embryos/eggs × 100%) [2,5,6] and deformed embryo percentages were determined [6,17]. All individuals that showed macroscopically abnormal developmental abnormalities, such as malformation of the body, stunned body, shortened and curved tail, heart and yolk sac edema, etc., were categorized as abnormal larvae. The percentage of individual deformities was not determined due to the fact that in many individuals, they occurred together, and it was not possible to classify deformations in only one class.

#### 2.2.2. Design of Experiment No. 2

In experiment 2, eggs were collected from eight females, and milt was collected from seven males. The whole procedure was conducted as presented in the description of experiment 1. The following were tested: TW (tap water; control group), 0.7% NaCl solution, V, B1, and B2 (Table 2). All study groups were performed on T (tap) and deionized (D) water. The eggs were incubated until hatching in water at a temperature of 12.0 °C, and then hatching rate and deformed embryo percentage were determined as described above.

#### 2.2.3. Mass Fertilization in Field Conditions

Mass fertilization was performed at three field hatcheries. Two research variants, V and B1, were prepared with hatching water. In the control group, gamete activation was performed with hatching water. The eggs were divided into three groups weighing 1.0 kg, mixed with 5 mL of pooled milt, and activated with hatching water (HW, control group for each hatchery), and V and B1 were prepared with hatching water. After mechanically removing adhesiveness, the eggs were transferred to Weiss jars (8 L each), where they were incubated until hatching in water at a constant temperature of 12.0 °C. Hatching rate and deformed embryo percentages were then determined as described above.

### 2.3. Incubation Conditions

The hatchery incubation water parameters were as follows: dissolved oxygen > 7 ppm; ammonia < 0.1 ppm; nitrite < 0.1 ppm; conductivity (25 °C), 516 μS; pH, 7.43; total water hardness, 255 mg/L CaCO_3_; Na^+^, 10.8 mg L^−1^; K^+^, 4.9 mg L^−1^; Fe, 12 μg/L; Mn, 26 μg/L. The fertilized eggs were incubated in a recirculating system at 12.0 °C (±0.1), which is the optimal temperature for northern pike embryo development [44]. Oxygen saturation exceeded 90%. Ammonia and nitrite concentrations were measured twice daily [48], and these compounds were not detected during the experiments (Hach Lange GmbH, Düsseldorf, Germany). Embryo survival percentages were determined in the hatching stage of development (total number of hatched larvae/total number of eggs in the sample × 100%). Once completed, the hatching rate and abnormal larvae [5,6,17] percentages were determined based on the total number of eggs in the samples (number of hatched larvae with body deformities/total number of hatched larvae in the sample × 100%).

### 2.4. Statistical Analysis

The normality of data distribution, expressed as means ± SD, was determined with Statistica 13.1 (StatSoft, Inc., Tulsa, OK, USA). The data were normally distributed (Shapiro–Wilk test), and the variances were homogenous (Levene’s test). Differences in means among groups for each variable were assessed with ANOVA and Tukey’s multiple range tests for group comparisons (values were significant at *p* < 0.05).

## 3. Results

The results of experiment 1 are presented in Figure 1. In all cases, preparing activation media with T versus D water resulted in lower fertilization percentages and, consequently, lower hatching rate percentages (a few to more than 40% difference in survival). The highest percentage of hatched embryos was observed in groups of eggs fertilized with the saline diluent prepared with deionized water at 61.2% and 56.5% for groups NaCl5-D and NaCl6-D, respectively. The lowest hatching percentage was noted in gamete groups activated with Woynarovich solution at 9.7% and 3.8% in the V-D and V-T groups, respectively. Survival in the V-D group was similar to that in the control group, in which TW (tap water) was used to activate gametes. The abnormal larva percentages were low in all groups (>0.3%).

The results of experiment 2 are presented in Figure 2. In all cases, preparing activation media with T versus D water resulted in lower fertilization percentages and, consequently, lower hatching rate percentages (a few to more than 60% difference in survival). In this case, the highest percentage of hatched embryos was in the groups of eggs activated with Woynarovich solution at 75.5% and 74.7% for V-D and V-T, respectively. The hatching percentage observed in most of the experimental groups was fairly even and ranged from 52.9% to 65.7% in NaCl7-D and B2-D, respectively. In the second experiment, the survival rate in the worst research group (B1-T) was more than five times higher than the survival rate obtained in the control group, in which TW (tap water) was used to activate gametes. The lowest hatching percentage was observed in groups of eggs activated with B1-T (26.1%) and T water (4.89%). The percentage rates of abnormal larvae were low in all groups (>0.3%).

The results of the field tests varied considerably and were completely different for each hatchery. In this study on the mass fertilization of eggs of northern pike in field hatcheries, the highest hatching rate compared to the control (means using hatchery water for gamete activating—one hatchery) or higher than the control (two hatcheries) was obtained in group B1 (Table 3). The highest survival rate was that of group V at the hatchery in Sterławki; at the Pasłęk hatchery, the survival rate was 0%. The percentage of abnormal larvae was low and did not exceed 0.5%.

## 4. Discussion

In this study, various activation media were tested simultaneously, and their influence on fertilization efficiency, expressed in this paper as hatching rate, was analyzed. Sperm extenders have been used for many decades in controlled pike breeding. As early as the 1970s, Billard [45,49] developed D532 diluent, which was intended for rainbow trout but can also be used successfully in pike breeding [50]. Later, the formulation of D532 was modified slightly [47]. It has also been proposed to use a NaCl saline medium to fertilize eggs instead of lake or T water [40,51]. Although it would seem that carrying out fertilization in the water in which spawners live should have the highest value of fertilization and survival of embryos, in hatching conditions, the use of media activating results in obtaining higher values. This is probably due to the prolonged sperm movement time and the delay in micropyle closure (e.g., [7,10,24,51,52,53]). It is also possible to use Woynarovich solution, which is commonly used during the controlled reproduction of wild and cultured cyprinids [7,17,28,54,55,56]. The use of such extenders always produces better results in terms of the percentages of developing eggs and hatching eggs compared to fertilizing eggs in water. It is important to select diluents that permit repeatedly attainment of the highest percentage of developing eggs and one that is made from as simple a recipe as possible and that is inexpensive to prepare.

The hatching percentages in the experimental groups in which the gametes were activated with any sperm diluent were higher than those noted in the experimental groups of gametes activated with T water. Other authors regularly report similar findings for pike [41] and other fish species, such as rainbow trout [25]. However, during our previous experiments on northern pike gametes, we never observed differences as great as those noted in the present study [43,44,57]. This is also unlikely to be linked with the osmolality of T water because sperm diluents were also prepared with this type of water. The quality of the egg pool used in experiment 2 was good based on routine microscopic observations of sub-batches. This phenomenon should be studied further in the future.

In most cases, the hatching percentages in groups of gametes activated with diluents prepared with D water were higher than those in groups activated with diluents prepared with T water, but this was not the rule, and the differences were not statistically significant (V-D and V-T in experiment 2). This aspect also requires further study and explanation. Interestingly, very divergent—even extreme—results were obtained on fertilization efficiency and embryo survival after applying V in experiments 1 and 2 (Figure 1 and Figure 2) and during field tests (Table 3). This indicates the need to conduct extended studies in the future, simultaneously recording sperm motility and duration. Using sperm diluents, such as Billard’s D532 solution, generally prevents egg adhesiveness. A few minutes following gamete activation, the egg batches were rinsed with hatchery water and transferred to incubators, where no further adhesiveness was observed. The results of applying experimental research work on semitechnical scales or in field hatcheries are sometimes slightly different (e.g., [10,28]), which was the case in this study, in which significant variation in results when using solution V as a gamete activation medium was confirmed. Scientific research, even if revealing, cannot always be immediately applied to breeding of animals (e.g., [58]). Therefore, if the test results can be practically and immediately applied to practice—in this case, for hatchery practice in aquaculture—it positively influences the development of aquaculture (e.g., [59,60]) and introduces measurable economic effects (e.g., [23]). In these studies, the use of activating media has been shown to have the effect of increasing fertilization and hatching rates. However, as shown by the results presented in Table 3, during the field tests, the selection of the activating medium for each hatchery should be done on a case-by-case basis, especially when hatchery or lake (river) water must be soluble for chemicals.

The highest percentage of motile northern pike sperm was noted in media with osmolalities between 125 and 235 mOsmol kg^−1^, whereas an osmolality exceeding 375 mOsmol kg^−1^ inhibited sperm motility [32]. The osmolality of activation solutions affects both sperm motility percentages and the speed of their movement. By using a diluent with an osmotic pressure and pH that is optimal for a given species, it is possible to significantly increase the duration of the progressive movement of sperm and thus the percentage of eggs fertilized. A properly selected diluent can also have a positive effect on eggs. This was observed in experiment 2, where, in the group of water-activated gametes, a dramatically low hatching rate (4.89%) was noted, whereas in some groups activated with diluents, the noted hatching rates were very high (B2-D, 62.7%; V-D, 75.55%; and V-T, 74.55%). When eggs are over-ripe or collected without due care, damage can occur, and egg contents—mainly yolk—can contaminate entire batches. After adding water, the yolk precipitates and agglutinates the sperm, thus hindering movement [25]. In this case, a decrease in pH is also observed, which has a detrimental effect on sperm motility [28,37,38,61]. Additionally, any yolk released clogs micropyles, which prevents eggs from being fertilized. Using diluents helps to avoid these problems. Even if the diluent is not buffered to the optimal pH, neutralizing problems caused by the presence of yolk from damaged eggs facilitates much higher egg fertilization percentages than when gametes are activated with water.

## 5. Conclusions

In this study, various activation media were tested for insemination of northern pike gametes. The results show that the effectiveness of various media depends largely on the type of water (tap, hatchery, or deionized) in which the chemicals are dissolved. This has been demonstrated in both experimental and field studies. Taking into account that the B1 activation medium, apart from a high fertilization percentage, also reduces egg adhesion, it is recommended for use in field hatcheries. A very large variability of the results obtained after using the V medium was also found. Therefore, it seems reasonable that each field hatchery should choose the medium that, under its own conditions, achieves the highest embryo survival and hatching rates.

## Figures and Tables

**Figure 1 animals-12-01022-f001:**
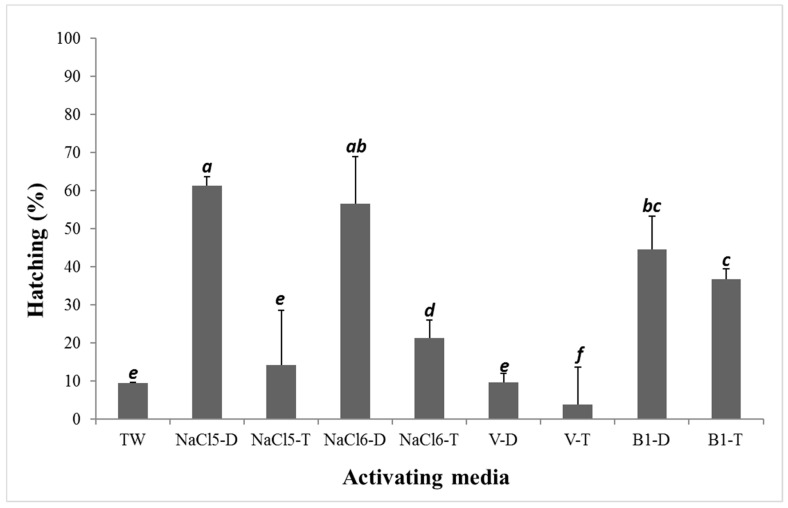
Hatching rates (%) of northern pike (*Esox lucius*) embryos in different activation media used for fertilization in experiment 1. Data are means ± SD. Values with different letter indices differ significantly (Tukey’s multiple range tests at *p* < 0.05).

**Figure 2 animals-12-01022-f002:**
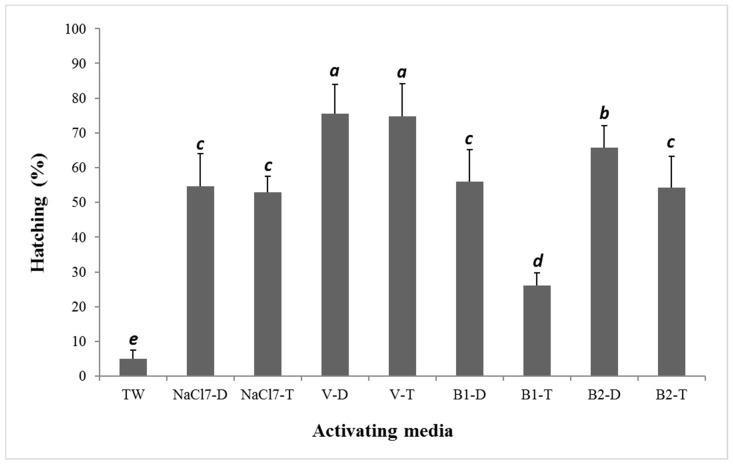
Hatching rates (%) of northern pike (*Esox lucius*) embryos in different activation media used for fertilization in experiment 2. Data are means ± SD. Values with different letter indices differ significantly (Tukey’s multiple range tests at *p* < 0.05).

**Table 1 animals-12-01022-t001:** The composition and pH of different activating media used in Experiment No. 1 for northern pike (*Esox lucius*) gametes. pH was measured during these studies. The “Author” column presents the authors of the work, the year of publication, and the reference number.

Medium	Composition of Diluent	pH	Author
TW	Tap water (T)	7.35	
DW	Deionized water (D)	6.37	
NaCl5-D	86 mM NaCl, water D	6.05	
NaCl5-T	86 mM NaCl, water T	7.30	
NaCl6-D	103 mM NaCl, water D	6.07	
NaCl6-T	103 mM NaCl, water T	7.27	
B1-D	94 mM NaCl, 20 mM Tris, 50 mM glicine, water D	8.73	Billard (1977) [44]
B1-T	94 mM NaCl, 20 mM Tris, 50 mM glicine, water T	8.62	
V-D	68 mM NaCl, 50 mM urea, water D	5.55	Woynarowich and Woynarowich (1980) [45]
V-T	68 mM NaCl, 50 urea, water T	7.00	

**Table 2 animals-12-01022-t002:** The composition and pH of different activating media in Experiment No. 2 for northern pike (*Esox lucius*) gametes. pH was measured during these studies. The “Author” column presents the authors of the work, the year of publication, and the reference number.

Medium	Composition of Diluent	pH	Author
TW	Tap water (T)	7.35	
DW	Deionized water (D)	6.37	
NaCl7-D	120 mM NaCl, water D	5.40	Babiak et al. (1997) [40]
NaCl7-T	120 mM NaCl, water T	7.00	
B2-D	125 mM NaCl, 20 mM TRIS, 30 mM glicine, 1 mM CaCl_2_, water D	8.86	Billard (1986) [46]
B2-T	125 mM NaCl, 20 mM TRIS, 30 mM glicine, 1 mM CaCl_2_, water T	8.73	
B1-D	94 mM NaCl, 20 mM Tris, 50 mM glicine, water D	8.73	Billard (1977) [44]
B1-T	94 mM NaCl, 20 mM Tris, 50 mM glicine, water T	8.62	
V-D	68 mM NaCl, 50 mM urea, water D	5.55	Woynarowich and Woynarowich (1980) [45]
V-T	68 mM NaCl, 50 mM urea, water T	7.00	

B2, second Billard solution.

**Table 3 animals-12-01022-t003:** Hatching rates with different activation media at different field hatcheries. Data are means ± SD. Values in columns with different letter indices differ significantly (Tukey’s multiple range tests at *p* < 0.05).

Hatchery/Group	Pasłęk	Pasym	Sterławki
HW	84.2 ± 2.2 ^a^	57.8 ± 3.3 ^b^	60.3 ± 2.9 ^b^
V	0.0	26.3 ± 3.6 ^c^	73.2 ± 2.2 ^a^
B1	84.5 ± 1.9 ^a^	72.9 ± 2.9 ^a^	64.1 ± 3.5 ^b^

## Data Availability

The data presented in this study are available on request from the corresponding author. The data are not publicly available due to privacy.

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
