# Peer review of "Effect of Application of Different Activation Media on Fertilization and Embryo Survival of Northern Pike, (Esox lucius) under Hatchery Conditions"

_animals, 2022, doi:10.3390/ani12081022_

Round 1

Reviewer 1 Report

To dear writers
I think it's an interesting manuscript. I reviewed it. It may be lacking in knowledge, but it was sometimes incomprehensible as a way to organize the discussion. Below is a list of the points I wonder.

1.Why is the experimental result not including DW alone?

2.How long is the storage period of "2.2.1" eggs and milts?

3.Show me the instrument that measured pH and conductivity.

4.The results of "V-D" and "V-T" differ greatly between Experiment 1 and Experiment 2. It should be clear and a little more detailed in the discussion. Since it is a freshwater fish, there is a high possibility that it will not hatch when the pH drops, but in "V-D", why is Experiment 1 low and Experiment 2 high? There is a little explanation, but I'm not sure.

5.It is easier to understand if the order of "V-D", "V-T", "B1-D", and "B1-T" is aligned in Fig. 1 and Fig. 2.

6.The alphabet above the bars in Figures 1 and 2 (a~ f) should be explained in Figure Legend.

7.In this discussion, the story of milk on line 225 is sudden. You should write in a little more detail to use it. Or, I think it should be used for the next experiment (report).

8.I think the water that used to raise the parent fish has a higher hatching rate than the water quality you indicate. I think this should also be considered.

9. What does it mean to separate Experiment 1 and Experiment 2? I didn't understand even after reading the methodology.

Author Response

The detailed answer, point by point, was presented in attachment.

Reviewer 2 Report

In my opinion, the article is interesting and novel, however it has many defects that should be corrected in order to be published.

Introduction

I believe that the justification for the interest in testing different dilution media should be addressed in a little more detail. Thus in the discussion (from lines 197 to 207), this concept is well described. It would be appropriate to do it in the introduction.

Objectives

I consider that the objective is excessively general, it could be maintained if the specific objectives of each of the experiments that are proposed to achieve them are cited. In fact, there are targets in the material and methods that should be removed from that section.

Material y methods

I do not think the general organization of material and methods is the clearest or most attractive for the reader.

I think it would improve greatly if it was organized by experiments.

Specific comments

Line 87 “ Journal of Laws of 2015, item 266, of January 15, 2015”…Please include the scope of law, EU???

Line 91. Describe in more detail the number of males and females, their weight, acclimatization time after capture, etc. Although in my opinion it would be clearer to use "Biological material or broodstock" and later specify this by experiment.

Line 100. In my opinion it would be important to indicate the location of each of the hatcheries.

Line 106. Please, indicate this in objectives, I think it is not appropriate in material and methods.

Line 108. Please, indicate the abbreviation of each diluent, for example “one with T”…rewrite as  “one with T (TW) ”.

Line 111. Please, specify the number of samples and their total (considering that they are done in triplicate).

Line 112. Detail microscopy equipment and procedure, please.

Line 121. Explain briefly how the selection of deformed embryos is carried out.

Table 1. In the author column put "this study" if applicable.  Check references format and also review the title of the table.

Table 2. In the author column put "this study" if applicable.  Check references format and also review the title of the table.

Line 127-135. Since the material and methods of experiment 1 and 2 are very similar, I think it would be appropriate to indicate 1 and refer only to the modifications in 2.

Line 138. These diluents have not been previously described... It is not understood why these were selected??

Line 147-156. It would be possible to include more detail: Volume of the containers, evaluation time of the hatching of the larvae, etc.

Results

I think the results are not described correctly; they must be independent of the figure. That is, the results must be described in detail.

Figure 1. Please rewrite the title of the figure so that it is independent of the text. Change “Fertilizing solution” per “diluent”.

Line 184. The hatchery control has not been previously defined in material and methods.

Discussion

In my opinion, it should be considered to write it again describing in a briefer way some items, such as the one mentioning to the 197-207 lines.

It would be interesting if the authors highlighted the interest of the tests carried out to try to obtain conclusions on the use of the different diluents in laboratory conditions versus companies.

Line 231-233. Results... must be included in the m&m and in results. It was not indicated anywhere that it was going to be evaluated. It must be taken into account in all the epigraphs of the work.

Conclusion

Please rewrite it to make it clearer and simpler.

References

I miss the review of the recent work https://doi.org/10.1016/j.aquaculture.2020.735575, in which some aspects evaluated in this study have also been addressed. It can even be used in an illustrative way in the way of presenting results in the same field.

Author Response

Dear Reviewer

Thanks you a lot for constructive comments. We corrected the MS according to your suggestions. The detailed answer you found in the additional file.

Reviewer 3 Report

This article with the title „ The effect of application of different activation media on fertilization and embryo survival of northern pike, Esox lucius, under hatchery conditions“ is written in easy-to-follow and consistent way. Methods and results are clearly stated. I highly appreciate that this study is focused on the freshwater fish specie and, moreover, on not commonly scientifically discussed specie. It is highly needed to publish studies on similar topics as they are not quite available. Therefore, I recommend this article for publication after some minor changes.

Title: appropriate

Simple summary: easy to follow

Abstract: Clear enough; however, I am missing the information about whether the pH of egges could be used as an quality indicator as claimed in simple summary, so please provide this info into the abstract

Keywords: sufficient

Introduction: Clearly indicates the problematics, easy to follow, appropriate literature sources.

70-72: please correct to appropriate common names according to, for example, Fishbase.org

M&M:

2.1 I am not familiar with the Poland laws, but if you evaluated also the larval stages – shouldn’t the document dealing with the experimental animals and the approval no. also be mentioned? Due to larvae is considered to be an animal in our country; however, embryo is not. Please, clarify this problematics for me.

I would divide the chapter “Experimental design” into two separate sub-chapters and give them together with the chapter 2.2.1 and 2.2.2., otherwise it is not good to firstly mentioned the origin and gametes collection for spawning and in the following chapter start to explain that there are two experiments – doesn’t bring clarity, so I recommend:

2.2 Experimental design – general content

2.2.1. Design of experiment no. 1

2.2.2. -II- no.2

112 – how many eggs were examined in total? why only “small batches” and not all of them were individually examined? why eggs from 5 females were used, only?

136, 144 – “deformed embryo percentage were determined as described above“ – but it is not described, it is only referred – so shouldn’t it be described at least briefly?

Also, I am missing some definitions in the M&M, for example – what is considered as an “abnormal larva”? Nowhere is, in my opinion, given the information of how long the hatching in this fish specie lasting, i.e. how long the incubation lasted. Also, why didn’t you evaluated the embryos lethality according to, for example, OECD 236 methodology, where the apical observations are described? (i.e. coagulation of embryos, lack of somite formation, non-detachment of the tail, and lack of heartbeat).

Results

I am missing the result about the pH and whether it can be applied as an appropriate indicator of embryo quality, as claimed in summary.

Fig. 1+2: I am suggesting to sort data from the highest to the lowest % of successful hatching to make the figure clearer

170 – “differ significantly statistically“ – so many words, i believe that „differ significantly“ is enough

174-175, 186-187: “abnormal larva” – I am missing the definition

Discussion: The discussion is easy to follow, carry on from general to concrete problems and specific results.

Author Response

(The authors gave the same response as above.)

Round 2

Reviewer 1 Report

Dear author, thank you for your answers regarding my review. thank you very much. But I still don't understand Question 6. I think many readers are wondering.

6.The alphabet above the bars in Figures 1 and 2 (a~ f) should be explained in Figure Legend.

Reviewer 2 Report

Thank you very much for the quick and efficient correction of the suggestions given by the reviewers. I consider that the work has been much better in its new version.

As a last suggestion, I consider that the information contained in the lines (91-93) is not required, I would eliminate it.

Best regards